# The Inhibitory Effect of Celangulin V on the ATP Hydrolytic Activity of the Complex of V-ATPase Subunits A and B in the Midgut of *Mythimna separata*

**DOI:** 10.3390/toxins11020130

**Published:** 2019-02-22

**Authors:** Liwen Ding, Zongxin Guo, Hang Xu, Tie Li, Yuanyuan Wang, Hu Tao

**Affiliations:** 1College of Chemistry and Pharmacy, Northwest A&F University, Yangling 712100, China; dingliwen@nwsuaf.edu.cn (L.D.); guozongxin1997@nwsuaf.edu.cn (Z.G.); xuh@nwsuaf.edu.cn (H.X.); 2010013814@nwsuaf.edu.cn (T.L.); 2College of Life Sciences, Northwest A&F University, Yangling 712100, China; yywangd@nwsuaf.edu.cn

**Keywords:** Celangulin V, *Mythimna separata* Walker, V-ATPase AB subunits complex, ATP hydrolysis

## Abstract

Celangulin V (CV) is a compound isolated from *Celastrus angulatus* Max that has a toxic activity against agricultural insect pests. CV can bind to subunits a, H, and B of the vacuolar ATPase (V-ATPase) in the midgut epithelial cells of insects. However, the mechanism of action of CV is still unclear. In this study, the soluble complex of the V-ATPase A subunit mutant TSCA which avoids the feedback inhibition by the hydrolysate ADP and V-ATPase B subunit were obtained and then purified using affinity chromatography. The H^+^K^+^-ATPase activity of the complex and the inhibitory activity of CV on ATP hydrolysis were determined. The results suggest that CV inhibits the ATP hydrolysis, resulting in an insecticidal effect. Additionally, the homology modeling of the AB complex and molecular docking results indicate that CV can competitively bind to the AB complex at the ATP binding site, which inhibits ATP hydrolysis. These findings suggest that the AB subunits complex is one of the potential targets for CV and is important for understanding the mechanism of interaction between CV and V-ATPase.

## 1. Introduction

*Celastrus angulatus* Maxim is China’s traditional insecticidal plant and is widely distributed in the Yellow River and Yangtze River basins [1]. Celangulin V (CV) is a natural product isolated from the plant and has been found to act on the midgut cells of the oriental armyworm larvae by immunoelectron microscopy [2,3]. It mainly acts on the plasma membrane and the protective layer of the cuticle on the gut cells known as the intima. The action of CV on the armyworm muscle cells results in obvious lesions of organelles such as mitochondrial swelling; an incomplete bilayer membrane; cytoplasm density reduction; arranged organelles disorder; and endoplasmic reticulum dilation [4]. An analysis of the digestive enzyme activity showed that there was no significant change in the activities of protease, amylase, or lipase in the midgut of poisoned insects, compared with normal insects. This suggests that CV mainly acts on the plasma membrane of the midgut cell and its intima system [5]. Further studies using affinity chromatography demonstrated that CV might bind to subunits a, H, and B of the armyworm *Mythimna separata* Walker (*M. separata*) midgut vacuolar ATPase (V-ATPase) [6]. Moreover, the H subunit was proved to be one of the potential targets of CV by a binding study between CV and the H subunit [7]. Nevertheless, the mechanism of how the binding of CV to the H subunit affects the holo V-ATPase activities, including proton pump activity and ATP hydrolysis, is not yet clear. One hypothesis suggested that the CV could bind well in the interdomain cleft of subunit H and make the ligand–protein complex more stable [8]. Because subunit H is located on the outer side of the V-ATPase complex, it is unclear how the more stable CV–H subunit complex affects the activities of V-ATPase. Meanwhile, the poor correlation between the binding of CV analogs to subunit H and the insecticidal activities of CV analogs hints that the V-ATPase complex contains multiple binding sites of CV [7].

V-ATPase is widely distributed in many organisms and participates in various physiological processes. V-ATPase in eukaryotes mainly acts as a proton pump and participates in the acidification of cellular compartments [9]. V-ATPase consists of two major components, the V_0_ sector for proton translocation and the V_1_ sector for ATP hydrolysis. The V-ATPase is comprised of fourteen different subunits, A_3_B_3_CFE_3_FG_3_Hac_x_c'_y_c''_z_de, in which unknown numbers of c, c', and c'' subunits are represented by stoichiometry x, y, z [10]. Most V-ATPase subunits are necessary for a functional V_0_V_1_ complex in vivo. The A_3_B_3_ hexamer formed by the A subunit alternating with the B subunit is the core part of the V_1_ complex, which is the catalytic center of ATP hydrolysis. The proton pump V_0_ complex is driven by the hydrolysis of ATP. Because ATP hydrolysis activity is feedback-inhibited by ADP, it is difficult to directly determine ATP hydrolysis activity in vitro. A previous study on the A subunit of *Thermococcus thermophilus* (*T. thermophilus*) revealed that the mutation of serine 232 to alanine and threonine 235 to serine of the A subunit can reduce ADP feedback inhibition without affecting ATP hydrolysis activity [11]. Thus, the complex of subunits A and B can be reconstructed in vitro, and its ATP hydrolysis activity can be measured. The mutant TSCA (C254A and T257S) was designed by comparing the homology modeling structure of the A subunit of *M. separata* with the structure of the A subunit of *T. thermophilus*. In this study, the TSCA and B subunits were expressed in *E. coli*. The inclusion bodies obtained by prokaryotic expression were refolded to obtain the soluble TSCA and B subunits complex. The ATP hydrolysis activity of the TSCA–B complex and the inhibition of CV to ATP hydrolysis were measured, and the binding of CV was further explored and proved by computer-aided molecular simulation.

## 2. Results

### 2.1. Expression, Refolding, and Purification of the Complex of the Recombinant Proteins

The TSCA and B subunits were inclusion bodies when they were expressed in *E. coli*. The inclusion bodies were solubilized in denaturing buffer at a 1:1 molar ratio and then refolded by refolding buffer. Most of the proteins were found in the supernatant. The renatured sample was passed through a nickel column. The desired proteins complex was obtained and the molecular weight (MW) was consistent with the theoretical value (Figure 1). 

### 2.2. ATP Hydrolysis Activity Assay of the TSCA–B Complex

The purified TSCA and B subunit complex was assayed for ATP hydrolysis activity using distilled water as a control, and the test was repeated three times. The ATP hydrolysis activity was calculated and processed as the Lineweaver-Burk of V–[S] curve, as shown in Figure 2a. The linear regression equation obtained *y* = 0.4819*x* + 1.792 was calculated to a Km value of 268 μM.

### 2.3. The Inhibitory Activity of CV to ATP Hydrolysis

The effect of CV on the ATP hydrolysis activity of the TSCA–B complex is shown in Figure 2b. The ATP hydrolysis activity of the TSCA–B complex decreased with the increase of CV concentration, supporting a competitively inhibitory model. The IC_50_ was found to be 69.8 µM by plotting the experimental data using the sigmoidal dose–response model of GraphPad Prism version 6.01 for Windows (La Jolla, CA, USA). Therefore, the Ki value of CV on the ATP hydrolysis activity of TSCA–B complex was calculated to be 10.0 µM.

### 2.4. Molecular Simulation

Homology modeling and molecular docking were carried out to understand the mechanism of inhibition of ATP hydrolysis by CV. Selecting the crystal structure of V_1_-ATPase from *T. thermophilus* as a template (Figure 3a, PDB ID: 3w3a [12]), the structure of the AB complex from *M. separata* was generated with SPRING server [13] (Figure 3b). The sequences of the template and the target protein share a 50.3% identity and 90.5% of aligned residues, this resulted in the two structures merging well (Figure 3c). After rebuilding a full-atoms protein model from the above modeling structure by PULCHRA [14], the molecular dockings of CV and ATP to the AB complex were carried out with the AutoDock 4.2 software [15] (Figure 3d,e). The energy of CV binding with the AB complex was predicted to be −4.85 kcal/mol, and the energy of ATP binding with the AB complex was predicted to be −2.03 kcal/mol. The G414, K437 residues of subunit A and I348, T349 of subunit B directly interacted with CV (Figure 3f). On the other hand, D416, D436, K437, K438 of subunit A and T439 of subunit B were involved in the binding of ATP (Figure 3g). Both compounds competed on the binding of K437 to subunit A. Therefore, it is conceivable that both the binding energy and interaction model support the mechanism of competitive binding of CV and ATP to the AB complex.

## 3. Discussion

*C. angulatus* is a widely distributed insecticidal plant in China. Its insecticidal active ingredients are basically polyesters with a skeleton of dihydroagarofuran sesquiterpenoid [16]. Celangulin V is a representative compound of this class of natural products which predominantly acts on the plasma membrane and endomembrane system of insects’ intestinal midgut cells, causing perforation of the insectile midgut. The mechanism of CV affects the midgut cells by targeting the V-ATPase in the cell [5]. 

V-ATPases are mainly located in the plasma membrane [17]. In eukaryotic endo-membranes, V-ATPases acidify specific organelles, such as lysosomes and secretory vesicles [18]. V-ATPase is also involved in the maintenance of pH stability and membrane electrification in many animal cell plasma membranes [19]. For insects, in addition to acidifying specific intracellular organelles, V-ATPases also play a crucial role in cationic transport in epithelial cells such as the salivary glands, labial glands, and midgut epithelia, through the reverse transport of K^+^/H^+^ ions. The goblet cell V-ATPase of the midgut apical membrane of *Manduca sexta* was the first insect vacuolar ATPase found in the plasma membrane, and is responsible for the alkalinization of the intestinal lumen [20]. In regard to the important function of V-ATPase, inhibitors targeting V-ATPase have been widely reported and studied [21]. However, many inhibitors have been reported as targeting subunits “a” and “c” of the V_0_ sector [22]. CV was identified as a compound with a novel binding site to subunit H.

In the present study, the TSCA and B subunits of the V-ATPase of *M. separata* were cloned and expressed. The assembly of AB complexes requires ATP binding to help the protein fold correctly [11]. However, the ATP was removed after the refolding dialysis and affinity chromatography purification. The TSCA–B complex was shown to have ATP hydrolysis activity with a Km value of 268 μM. Because CV was hardly dissolved in the water and its soluble organic solvents such as DMSO and acetone affected the structure of TSCA–B complex causing the elimination of the ATP hydrolysis activity, the appropriate amount of CV was suspended in water and incubated at room temperature for half an hour to ensure successful binding. The CV was a suspension rather than a solution due to its low solubility in water. Therefore, effective CV concentration may have been lower than reported. CV was identified as a competitive ATP hydrolysis inhibitor with a Ki of 10.0 µM. The results of molecular simulation performed later suggest that CV and ATP will bind to K437 of subunit A, which is in line with the experimental enzyme activity assay data. 

It has been proved that CV binds to the H subunit of the V-ATPase of *M. separata* [6]. However, the correlation between the binding of CV analogs to the H subunit and the insecticidal activities of CV analogs is poor [7]. Obviously, the binding and regulation of CV to other units of V-ATPase is worth investigating. The inhibition of ATP hydrolysis by CV can directly result in V-ATPase dysfunction and lead to physiological effects such as changing the high alkaline environment in the midgut, inducing the accumulation of K^+^ in cells, and influencing the amino acid transportation to the gut cells. Finally, the dysfunction of V-ATPase will lead to death due to the loss of body fluid. 

Furthermore, studies on the other units, especially the “a” subunit of V-ATPase, are necessary to identify or opt out of the CV regulating subunits and to comprehensively describe the mechanism of action of CV.

## 4. Materials and Methods 

### 4.1. Plasmid

The plasmids *pET-22b-VATPase-A* and *pET-22b-VATPase-B* were provided by the Institute of Pesticide, Northwest A&F University. The plasmid *pET-22b-VATPase-TSCA* contains the C254A and T257S mutant gene of *pET-22b-VATPase-A.* The TSCA gene was amplified with overlap PCR and inserted into the *pET-22b(+)* vector. The primers for overlap PCR were as follows: Forward primer: CGCATATGAGCAAAAAAGATCAGCTGAAGAAGATCG;Reverse mutation: GAGATGAC**GCT**CTTGCC**CGC**ACCGAAGG;Forward mutation: CCTTCGGT**GCG**GGCAAG**AGC**GTCATCTC;Reverse primer: GCAAGCTTGTCCTCCAGGTTACGGAAGG. 

The underlined sequences are the restriction enzyme Nde I and Hind III cutting sites. The bold font sequences are the mutant sites.

### 4.2. Chemicals

The CV was a kind gift from Prof. Jiwen Zhang of Northwest A&F University; the nuclease was purchased from TaKaRa (Kyoto, Japan); the protein marker was purchased from Thermo Fisher Scientific (MA, USA); the Bradford kit was purchased from Tiangen (Beijing, China); the Ni-NTA agar was purchased from GE Healthcare (Pittsburgh, PA, USA); and all other chemicals were purchased from SanLi (Yangling, China).

### 4.3. Expression of TSCA and B Subunits of V-ATPase 

The recombinant plasmids *pET-22b-VATPase-TSCA* and *pET-22b-VATPase-B* were cultured in LB liquid medium at 37 °C. When the OD_600_ of the culture medium reached 0.6, isopropyl thiogalactoside (IPTG) was added at a final concentration of 1 mmol/L. The cells were induced at 20 °C for 12 h. The cells collected by centrifugation were resuspended in lysis buffer (50 mmol/L Tris-HCl, 150 mmol/L NaCl, pH 7.0). Then, nuclease (1:10^4^, *v/v*) and 1 mmol/L phenylmethylsulfonyl fluoride (PMSF) was added to the buffer. After being sonicated for 30 min (200 W, 5 s on, 5 s off), the solution was centrifuged at 4 °C for 30 min at 12,000 r/min. The supernatant and the precipitate were detected by SDS-PAGE. The results showed that most of the above proteins existed as inclusion bodies.

### 4.4. Denaturation and Renaturation of Inclusion Bodies

Because both the TSCA and B subunit proteins obtained were inclusion bodies, they were washed with 2% Triton X-100 wash buffer (50 mmol/L Tris-HCl, 150 mmol/L NaCl, pH 8.0), separately. They were then washed with 1% Triton X-100 wash buffer (50 mmol/L Tris-HCl, 150 mmol/L NaCl, pH 8.0). Finally, they were washed twice with Triton X-100-free wash buffer (50 mmol/L Tris-HCl, 150 mmol/L NaCl, pH 8.0). Each wash was accompanied by the grinding of suspension solution with a homogenizer for 10 min followed by centrifugation at 12,000 r/min, 4 °C for 20 min. The washing effects were identified by SDS-PAGE. The washed inclusion bodies were added to the denaturing buffer (8 mol/L urea, 1 mmol/L dithiothreitol, 50 mmol/L Tris-HCl, 150 mmol/L NaCl, pH 8.0) and stirred at room temperature for 4 h. The dissolved inclusion bodies were centrifuged at 12,000 r/min, 4 °C for 30 min. The supernatant was retained for SDS-PAGE. Finally, the Bradford method was applied to measure protein concentrations [23].

After the TSCA and B subunit inclusion bodies were dissolved, they were diluted with denaturing buffers, and the diluted TSCA and B subunit proteins were mixed at 1:1 molar ratio in a dialysis bag and then ATP was also added at 1:1 molar ratio. Dialysis was carried out for 12 h at 4 °C against refolding buffer (4 mol/L urea, 1 mol/L L-arginine, 1 mmol/L benzamidine, 50 mmol/L Tris-HCl, 150 mmol/L NaCl, 1 mmol/L dithiothreitol, pH 8.0). The urea concentrations were reduced in turn until the refolding buffer became urea-free at the end of the dialysis. Then, the dialyzed sample was purified by a nickel column with affinity chromatography.

### 4.5. ATP Hydrolysis Activity Assays of TSCA–B Complex

The in vitro activity of the ATP hydrolysis was determined by assay of the inorganic phosphorus content hydrolyzed. Two sets of reaction groups were prepared: the assay group, and the control group. Each reaction system contained 400 µL 50 mmol/L Tris-HCl (pH 7.0), 5 mmol/L MgCl_2_. Then, 100 µL of recombinant TSCA–B complex solution and 100 µL ddH_2_O were added to the assay group and the control group, respectively. The two groups of reactants were mixed well and incubated at 37 °C for 5 min. Subsequently, the ATP was added until its final concentration to 1.6 mM. After the mixture was incubated again for 5 min, the reaction was stopped by the addition of ice-cold trichloroacetic acid (15% *w/v*). Then, 50 µL of reagent (1% NH_4_Mo_7_O_24_·4H_2_O, 0.5% NH_2_SO_4_) and 100 µL of 1% ascorbic acid were added. After the reaction lasted for 25 min, the UV absorbance of the sample was determined at 600 nm. The concentration of the substrate ATP was changed in different assay groups so that the final concentrations of ATP in the reaction solution were 0.1, 0.2, 0.4, 0.8, and 1.6 mM, respectively. The experiment was repeated three times for averaging. Calculated as Equation (1), the enzyme activity indicated that the ATPase hydrolyzes ATP per milligram of the sample to produce 1 µmol of Pi in an hour (µmol Pi·mg^−1^·h^−1^). The double reciprocal plot was used to calculate the Km value with the linear regression model of GraphPad Prism version 6.01 for Windows (La Jolla, CA, USA).
(1)$$\begin{array}{l} \mathrm{V-ATPase}\;\mathrm{activity}=\frac{\mathrm{OD}\;\mathrm{of}\;\mathrm{the}\;\mathrm{determination}\;\mathrm{group}\;-\;\mathrm{OD}\;\mathrm{of}\;\mathrm{the}\;\mathrm{control}\;\mathrm{group}\;}{\mathrm{OD}\;\mathrm{of}\;\mathrm{the}\;\mathrm{standard}\;\mathrm{group}}\times \mathrm{standard}\;\mathrm{sample}\;\\ \mathrm{concentration}\left(0.5\;\mu m/\mathrm{mL}\right)\times \mathrm{sample}\;\mathrm{dilution}\;\mathrm{factor}\times \mathrm{reaction}\;\mathrm{time}÷\mathrm{protein}\;\mathrm{content}\left(\mathrm{mg}/\mathrm{mL}\right) \end{array}$$

### 4.6. Inhibitory Activity Assays of CV

Different CV samples were suspended in 100 µL of ddH_2_O and were added to the TSCA–B complex solutions which were at the same concentration. The final concentrations of CV in the reaction solution were 0.125, 0.25, 0.5, and 1 mmol/L, respectively. An equal volume of distilled water was added to the enzyme solution as a control. Since CV is hardly soluble in water, the sample was incubated at 37 °C on a constant temperature shaker at 50 rpm for 30 min to allow sufficient solubilizing and binding of CV. The residual enzyme activity was then determined by the above method using 1.6 mM ATP as the substrate. Each concentration of CV was measured three times for averaging. According to Equations (2) and (3), when the inhibitor concentration is zero, V_0_ = 1/b and the inhibitory concentration is b/a, the reaction rate is half of the maximum rate. Thus, at that time the concentration of the inhibitor is equal to b/a it is usually annotated as IC_50_. The IC_50_ was obtained by plotting the experimental data using the sigmoidal dose–response model of GraphPad Prism 6.01. Then, the Ki value was calculated as Equation (4) below:(2)$$a=\frac{Km}{\mathrm{KiVmax}[S]}$$
(3)$$b=\frac{\mathrm{Km}+[S]}{\mathrm{Vmax}[S]}$$
(4)$$\mathrm{Ki}=\frac{\mathrm{bKm}}{a(\mathrm{Km}+[S])}$$

### 4.7. Molecular Simulation

A homology model structure of the V-ATPase subunits AB complex of *M. separata* was built using SPRING server [13]. The crystal structure of V_1_-ATPase from *T. thermophilus* (PDB ID: 3w3a [12]) was chosen as a template. The structure obtained from the homology modeling only contained the carbon atoms and hydrogen atoms of the backbone. PULCHRA [14] was used for reconstruction of the full-atoms protein model. The studies of molecular docking were performed with the AutoDock 4.2 software [15]. Docking calculations were carried with the Lamarckian genetic algorithm (LGA). The docking parameters consisted of setting the population size to 150, the number of generations to 270,000, the number of evaluations to 25,000,000, the number of docking runs to 10, the number of top individuals that automatically survive to 1, the rate of gene mutation to 0.02, and the rate of crossover to 0.8. PYMOL [24] was used to display the results of molecular docking.

## Figures and Tables

**Figure 1 toxins-11-00130-f001:**
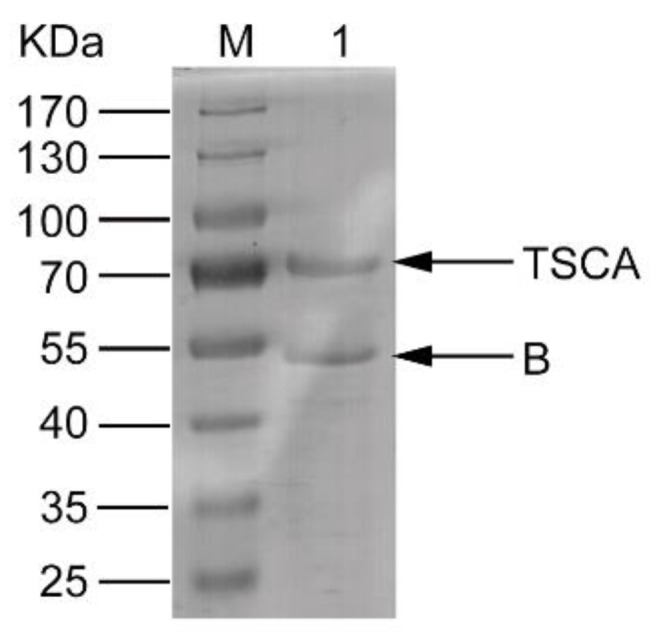
SDS-PAGE analysis of the TSCA–B complex purified by Ni-NTA column. Lane 1 presents the result of SDS-PAGE analysis of the eluent after Ni-NTA column purification. Lane M is a protein molecular weight (MW) marker (Thermo #26616).

**Figure 2 toxins-11-00130-f002:**
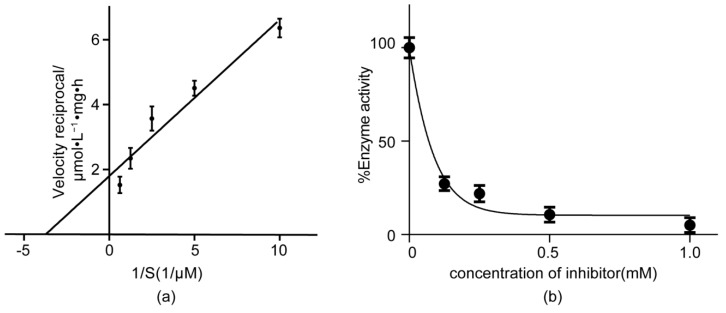
(**a**) Lineweaver–Burk of V–[S] curve analysis of the ATP hydrolysis activity of the TSCA–B complex. (**b**)Suppression curve of Celangulin V (CV) on the ATP hydrolysis activity of TSCA–B complex. The error bars represent SD.

**Figure 3 toxins-11-00130-f003:**
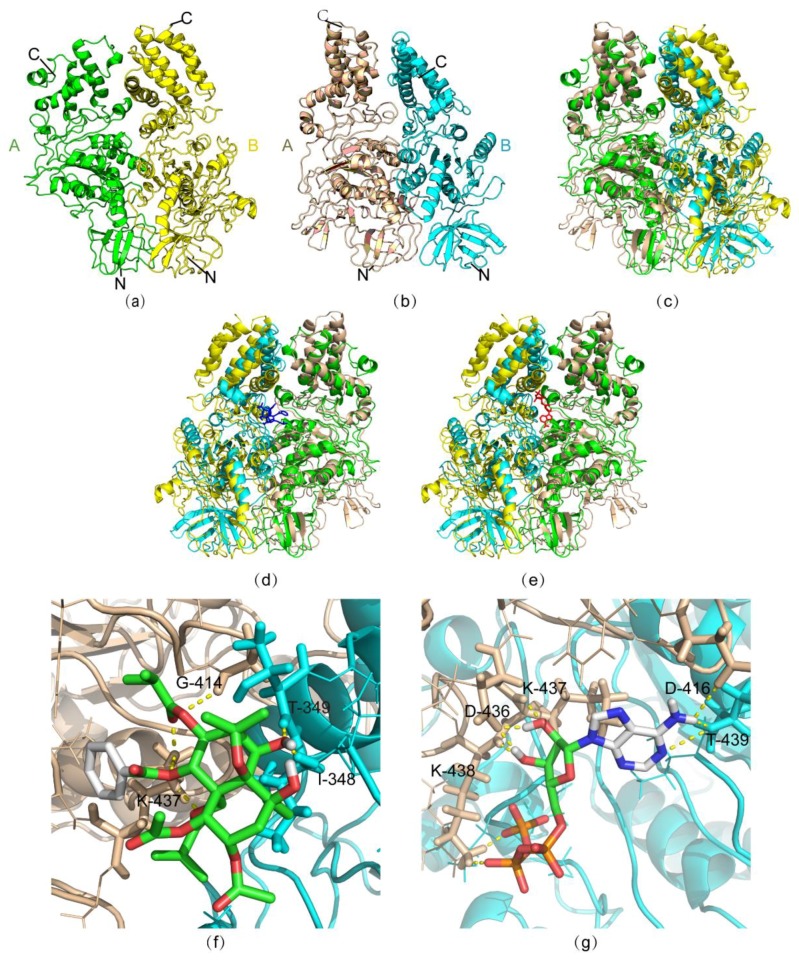
The homology modeling of the AB subunits complex of V-ATPase from *M. separata* and the docking of CV and ATP to the molecular model. (**a**) The structures of the *T. thermophilus* subunit AB complex as a template; subunit A is shown in green, and subunit B is shown in yellow. (**b**) The results of protein modeling of the *M. separata* subunit AB complex; subunit A is shown in wheat, and subunit B is shown in cyan. (**c**) Structural alignment of two different subunit AB complexes shown in (a) and (b). (**d**), The results of the docking the subunit AB complex from *M. separata* and CV; the *T. thermophilus* subunit AB complex structure is superimposed onto the structure of the *M. separata* AB complex of V-ATPase; CV is colored blue. (**e**) The docking result of the *M. separata* AB complex and ATP; the structure of the *T. thermophilus* AB complex is superimposed onto the *M. separata* V-ATPase subunit AB complex; ATP is colored red. (**f**) Schematic representation of the docking the *M. separata* subunit AB complex and CV. (**g**) Schematic representation of the docking of the *M. separata* subunit AB complex and ATP.

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
