# Peer review of "The Inhibitory Effect of Celangulin V on the ATP Hydrolytic Activity of the Complex of V-ATPase Subunits A and B in the Midgut of Mythimna separata"

_toxins, 2019, doi:10.3390/toxins11020130_

Round 1

Reviewer 1 Report

In order to study the mechanism of the small insecticidal molecule celangulin V (CV), the authors prepared a model of the M. separataV-ATPase V1complex using the A and B subunits that together carry out ATP hydrolysis. In order to study this reaction in vitro, they first had to mutate two amino acid residues of the A subunit to render it resistant to feedback inhibition by ADP. The two recombinant proteins were expressed in E. coli, then purified and renatured. The identity of the proteins was confirmed by MW on SDS-PAGE. 

The authors determined the Kmvalue for the TSCA-B complex in vitro. It would be interesting to know the Kmvalue for this complex in the presence of CV, as this should determine the mode of inhibition (competitive, non-competitive, etc.). Since this issue is raised later in the paper as a hypothesis based on computer modeling, it seems logical to carry out this experiment to verify the modeling results. 

Figure 2b shows an inhibition curve of CV on the TSCA-B complex in vitro. Unfortunately, the tested concentrations of CV are all well under 50% enzyme activity, leading effectively to extrapolation of the IC50value, which may hence be inaccurate. It would be better to repeat this experiment with a larger range of concentrations of CV so that, ideally, two concentrations would lie above 50% enzyme activity (i.e. lower concentrations of CV). The IC50was calculated with a sigmoidal regression, which is appropriate, but the graph is clearly not sigmoidal in shape, illustrating the fact that important data points are missing. The best that can be concluded from this data is that the IC50is <0.125 mM. Strangely, the IC50was calculated at 0.765 mM, which, as I read it, does not agree with the graph. The IC50 and Ki values need to be recalculated after re-doing the dose-response experiment with a larger range of concentrations, including some lower concentrations, to bracket the 50% activity value. 

The authors conclude (lines 147-148) that the results of the simulation “demonstrate CV and ATP will bind to K437 of subunit A.” Since these models are based on a hypothetical protein structure, the language should be more guarded: The results suggestthat CV and ATP… 

The authors state (lines 148-149) that the finding above is in accord with the experimental data. They should clarify which data they are referring to. 

Other notes:

The species name M. separataappears to have been auto-corrected to “separate” in many places in the text. 

In the introduction (lines 28-31), there are a number of confusing phrases (“intima of the midgut cells,” “fluff mess reduction,” “bilayer membrane uncompleted”) that seem to refer to citation #4. These phrases need to be clarified. 

It should also be mentioned that the cited paper (#4) was investigating muscle cells—not midgut cells. 

On line 127, Citation #17 does not mention the intima. (I may be wrong, but I think the intima is only found in the foregut and hindgut, not the midgut.) (See also line 28.)

Lines 156-157 also mention “fluff mess reduction”, needs to be clarified

Line 212 refers to “protease suspension.” Is this the solution containing the TSCA-B complex? The authors should clarify. 

Author Response

Response to Reviewer 1 Comments

Point 1: Figure 2b shows an inhibition curve of CV on the TSCA-B complex in vitro. Unfortunately, the tested concentrations of CV are all well under 50% enzyme activity, leading effectively to extrapolation of the IC50 value, which may hence be inaccurate. It would be better to repeat this experiment with a larger range of concentrations of CV so that, ideally, two concentrations would lie above 50% enzyme activity (i.e. lower concentrations of CV). The IC50 was calculated with a sigmoidal regression, which is appropriate, but the graph is clearly not sigmoidal in shape, illustrating the fact that important data points are missing. The best that can be concluded from this data is that the IC50 is <0.125 mM. Strangely, the IC50 was calculated at 0.765 mM, which, as I read it, does not agree with the graph. The IC50 and Ki values need to be recalculated after re-doing the dose-response experiment with a larger range of concentrations, including some lower concentrations, to bracket the 50% activity value.

Response 1: Thanks for the valuable suggestions. Because CV was hardly dissolved in the water and its soluble organic solvents such as DMSO and acetone affected the structure of TSCA-B complex causing the elimination of the ATP hydrolysis activity, the appropriate amount of DRY POWDER of CV was suspended in water and incubated with the TSCA-B sample solution to make sure them binding well. CV is hard to be weighted accurately when we try to make the test concentration of CV below 0.125 mM.

We are sorry for the mistake in the calculation of the IC50 of CV. It should be 0.0698 mM and Ki should be 0.01004 mM. We have revised our paper.

Point 2: The authors conclude (lines 147-148) that the results of the simulation “demonstrate CV and ATP will bind to K437 of subunit A.” Since these models are based on a hypothetical protein structure, the language should be more guarded: The results suggest that CV and ATP…

Response 2: Very good points. Thanks! We revised the paper as above.

Point 3:The authors state (lines 148-149) that the finding above is in accord with the experimental data. They should clarify which data they are referring to.

Response 3: To be changed to: “which is in accord with the experimental enzyme activity assay data

Point 4: The species name M. separate appears to have been auto-corrected to “separate” in many places in the text.

Response 4: We checked again and corrected accordingly.

Point 5:In the introduction (lines 28-31), there are a number of confusing phrases (“intima of the midgut cells,” “fluff mess reduction,” “bilayer membrane uncompleted”) that seem to refer to citation #4. These phrases need to be clarified. It should also be mentioned that the cited paper (#4) was investigating muscle cells—not midgut cells.

Response 5: To be changed to:” It mainly acts on the plasma membrane and the protective layer of cuticle on the gut cells known as the intima. The action of CV on the armyworm muscle cells results in…”

Point 6: On line 127, Citation #17 does not mention the intima. (I may be wrong, but I think the intima is only found in the foregut and hindgut, not the midgut.) (See also line 28.)

Response 6: We deleted the:”intima”.

Point 7: Lines 156-157 also mention “fluff mess reduction”, needs to be clarified

Response 7: We deleted the “fluff mess reduction”.

Point 8: Line 212 refers to “protease suspension.” Is this the solution containing the TSCA-B complex? The authors should clarify.

Response 8: To be changed to:” TSCA-B complex solution”.

Reviewer 2 Report

This manuscript on the inhibitory effect of Celangulin V to V-ATPase has some excellent data. The science is sound and their findings will be of interest to the CV field and toxinology in general. However, the findings are difficult to fully interpret due to some language issues. My main concern with this manuscript is that it needs to be thoroughly edited for language.

While as a reviewer my primary role is to evaluate the science and not to act as a language editor, I have listed below many examples of sections of the text that are difficult to read due to language. This list is not comprehensive.

7: Change to “the H, B, and A subunits…”

10: Change to “V-ATPase A subunit mutant TSCA which avoids the feedback inhibition…”

12: Change to “…the inhibitory activity of CV on ATP hydrolysis…”

13: Change to “…suggest that CV inhibits ATP hydrolysis…”

15: Change to “…ATP binding site, which inhibits ATP hydrolysis…”

16: Change to “…These findings suggest that the AB complex is one…”

25: Change to “…traditional insecticidal plant, widely distributed…”

28: The authors make reference to the “intima” multiple times throughout the manuscript, but they never define this particular aspect of insect physiology. They should define this when they first introduce this in the introduction. One suggestion would be: “It mainly acts on the plasma membrane and the protective layer of cuticle on the gut cells known as the intima. The action of CV results in…”

29: I have no idea what “fluff mess reduction” is, and the authors us this phrase here on line 29, and again on line 156. This needs to be clarified.

30: Change to “… incomplete bilayer membrane…”

32: Change to “…amylase and lipase in the midgut…”

35: Change to “…H, B, and A subunits…” There are several places where the authors fail to capitalize “A” when referring to the A subunit, this needs to be corrected.

41: Change to “…Because the H subunit is located on the outer side…”

43: Change to “… binding of CV analogues to subunit H and…”

70: Change to “Most of the proteins were found in the supernatant.”

71: Change to “The desired protein complex was obtained…”

95: change the species name to lower case (M. separate).

103: Change to “Both of the two compounds will compete to bind to K437…”

104: Change to “… it is conceivable that both the binding energy and …”

105: Change to “… of competitive binding of CV…”

123: Change to “… natural products that mainly acts on…”

125: Change to “CV effects the midgut cells by targeting the V-ATPase in cells.”

133: Change to “… membrane of H. gigas was the first insect vacuolar ATPase found in the plasma membrane, which is…”

136: Change to “… many inhibitors were reported to target subunits A and C of the V0 sector.”

137: Change to “…with a novel binding site to subunit H.”

142: Change to “…complex was shown to have ATP hydrolysis…”

146: Change to “… to make sure they were well bound. CV was identified…”

148: Change to “subunit A, which supports the experimental data…”

153: Change to “… worth investigating. The inhibition of ATP hydrolysis by CV can directly result in V-ATPase dysfunction and lead to physiological…”

156: Change to “… transportation to the gut cells.”

158: Change to “Future studies on the other units, especially the A subunit of V-ATPase…”

Author Response

Response to Reviewer 2 Comments

Point 1: This manuscript on the inhibitory effect of Celangulin V to V-ATPase has some excellent data. The science is sound and their findings will be of interest to the CV field and toxinology in general. However, the findings are difficult to fully interpret due to some language issues. My main concern with this manuscript is that it needs to be thoroughly edited for language.

Response 1:  We highly appreciate your constructive comments and have revised the manuscript accordingly. In addition, we have asked our friend, an American expert working in the same field, to improve the writing and substantially revise the manuscript.

Point 2: The authors make reference to the “intima” multiple times throughout the manuscript, but they never define this particular aspect of insect physiology. They should define this when they first introduce this in the introduction. One suggestion would be: “It mainly acts on the plasma membrane and the protective layer of cuticle on the gut cells known as the intima. The action of CV results in…”

Response 2: Very good points. Thanks! We revised the paper as above.

Point 3: I have no idea what “fluff mess reduction” is, and the authors use this phrase here on line 29, and again on line 156. This needs to be clarified..

Response 3: We deleted the fluff mess reduction” on line 29 and 156.

Round 2

Reviewer 1 Report

There are still instances where M. separata has been misspelled. 

The results for measuring IC50 should be further explained in the Results text: The CV was a suspension rather than a solution due to low solubility of CV in water. Therefore, the effective CV concentration may have been lower than reported. 

If possible, the CV suspensions should be diluted further, perhaps on a base 10 log scale, to include concentration values near 50% inhibition, and the inhibition experiment should be repeated. Since CV suffers from low solubility, I would not necessarily expect the inhibition curve to follow a simple sigmoidal shape. This only increases the need to include more data points at lower concentrations. As it stands, the calculated IC50 is only a wild guess. If the authors intend to accurately report an IC50 value, then this experiment needs to be repeated. If the IC50 is not of great importance to this study, then the experiment can be left as it is. But the limitations of the IC50 result must be carefully explained, or the IC50 result should be omitted altogether. 

Were the measurements in Fig. 2a and 2b from three independent experiments? If so, please indicate this in the Methods and report what the error bars represent (SD, SEM, or 95% CI). 

Author Response

Point 1: There are still instances where M. separata has been misspelled.

Response 1:  Thanks! We checked the manuscript and have revised it (line 4).

Point 2: The results for measuring IC50 should be further explained in the Results text: The CV was a suspension rather than a solution due to low solubility of CV in water. Therefore, the effective CV concentration may have been lower than reported.

Response 2: Thanks! We revised the paper and added the above text in line 143.

Point 3: If possible, the CV suspensions should be diluted further, perhaps on a base 10 log scale, to include concentration values near 50% inhibition, and the inhibition experiment should be repeated. Since CV suffers from low solubility, I would not necessarily expect the inhibition curve to follow a simple sigmoidal shape. This only increases the need to include more data points at lower concentrations. As it stands, the calculated IC50 is only a wild guess. If the authors intend to accurately report an IC50 value, then this experiment needs to be repeated. If the IC50 is not of great importance to this study, then the experiment can be left as it is. But the limitations of the IC50 result must be carefully explained, or the IC50 result should be omitted altogether.

Response 3: We tried incubating CV with TSCA-B complex for 5 min, 10 min, 15 min and 30 min. The inhibitory activities of CV were lower for incubating 5 min and 10 min. The activities for 15 min and 30 min incubation were almost the same and were higher than that of 5 min and 10 min. Meanwhile, we tried the temperature of incubation. The results showed no inhibitory activity while incubating at 4 . Therefore, we incubated at room temperature for half an hour.

Point 4: Were the measurements in Fig. 2a and 2b from three independent experiments? If so, please indicate this in the Methods and report what the error bars represent (SD, SEM, or 95% CI).

Response 4: Thank you! Yes, as shown in materials and methods, the data in Fig. 2a and 2b from three independent experiment. The error bars represent SD. We added it in line 90.